# MIST: Multiple Stochastic Prompt Tuning for Few-shot Adaptation under Extreme Domain Shift

## Abstract

Foundation Vision-Language Models (VLMs) like CLIP generalize well due to large-scale pretraining, but their performance degrades under significant distribution shifts in appearance and label semantics. Few-shot adaptation via adapter or prompt tuning addresses limited-data tasks, but are not specifically designed to handle such extreme domain shifts. Some cross-domain few-shot methods consider such domain-shifts but often use episodic settings with fixed classes, limiting real-world applicability. To address this gap, we propose a novel framework MIST (Multiple Stochastic Prompt Tuning), which adapts CLIP to extreme domain shifts with few labeled examples across all classes simultaneously. MIST uses multiple learnable prompts per class to capture diverse modes in visual features, modeled as Gaussian distributions to improve generalization and reduce overfitting. Extensive experiments show the effectiveness of the proposed framework.

## 1 Introduction

Foundation Vision-Language Models (VLMs) such as CLIP (Radford et al., 2021) and ALIGN (Jia et al., 2021) have advanced computer vision by enabling strong zero-shot generalization. Trained on large-scale image-text pairs, they learn robust representations transferable across tasks. However, their performance degrades on specialized or fine-grained tasks, where some task-specific adaptation is necessary. Since full pretraining is costly, there is growing interest in efficient adaptation techniques with few labeled examples from such tasks.

Few-shot adaptation of such models remains challenging due to overfitting and potential loss of pretrained generalization. Parameter-efficient tuning approaches, such as prompt tuning (Zhou et al., 2022) and adapter tuning (Gao et al., 2024), mitigate this by optimizing only a small set of parameters while keeping the backbone frozen. Despite these advances, these methods focus on standard benchmarks and overlook extreme domain shifts common in real-world applications, where datasets may differ greatly in visual appearance and label semantics — For example, medical image datasets often feature domain-specific content and class names that do not align with natural image concepts and are typically unavailable for pretraining due to privacy concerns. While recent works have applied CLIP to cross-domain few-shot learning (CDFSL) (Xiao et al., 2024), they typically adopt a source-free meta-testing setup, adapting to episodes with a few sampled classes (e.g., 5-way) from the target domain. Performance is averaged over many such episodes. However, this approach is computationally expensive and misaligned with real-world settings, where all target classes are present simultaneously.

In this work, we propose a novel prompt learning framework, *MIST* (Multiple Stochastic Prompt Tuning), for adapting CLIP to a more realistic setting, where target datasets exhibit significant domain and semantic shifts, and only a few labeled examples from all classes are available simultaneously. We first observe that extreme distribution shifts can lead to fragmented visual representations, forming separate and inconsistent clusters in the embedding space (Fig. 3). Moreover, occurrence of a large number of classes together with semantic shifts (different class labels) can cause multiple class features to cluster together, resulting in ambiguous decision boundaries. To address these challenges, we introduce multiple learnable prompts per class, enabling better modeling of multi-modal feature distributions. Further, instead of directly optimizing prompt weights, we represent each prompt as a Gaussian distribution with learnable mean and variance, promoting diverse and well-

separated representations while mitigating overfitting through efficient exploration of the prompt space. The key contributions of this work are summarized below:

1. We propose a novel framework for few-shot adaptation of CLIP to realistic scenarios with extreme domain and label semantic shifts, with all target classes present simultaneously.

2. We analyze limitations of existing prompt tuning methods for few-shot setting, under severe distribution shifts.

3. We propose *MIST*, a novel prompt tuning framework that uses multiple class-specific prompts to model multimodal visual feature distributions.

4. We further represent each prompt as a learnable Gaussian distribution, enabling better generalization and reducing overfitting in low-data regimes.

5. Extensive experiments demonstrate that MIST outperforms state-of-the-art methods across multiple challenging benchmarks.

## 2 RELATED WORK

Here, we briefly discuss the related work in literature.

**Vision-language foundation models.** Recently, Vision-Language Models (VLMs) (Radford et al., 2021; Jia et al., 2021; Li et al., 2022) have shown strong zero-shot generalization by learning aligned visual-textual representations from web-scale image-text pairs. However, their performance can degrade on domain-specific tasks not seen during pretraining, which has motivated recent works on efficient adaptation using limited labeled samples from the target domain.

**Few-shot adaptation of VLMs.** Adapting these large-scale models to downstream tasks with few labeled training data is often challenging, due to the risk of overfitting. Efficient transfer learning methods like prompt tuning (Zhou et al., 2022; Khattak et al., 2023a;b) or adapter tuning (Gao et al., 2024; Zhang et al., 2021) address this issue by optimizing only a few parameters added to these models, either in the input space or output layers. For instance, CLIP-Adapter (Gao et al., 2024) modifies visual features via a classifier, Tip-Adapter (Zhang et al., 2021) uses few-shot saved prototypes for guidance. CoOp (Zhou et al., 2022) trains prompt vectors added to the classname, keeping the CLIP encoders frozen. MaPLe (Khattak et al., 2023a) uses multimodal prompts in both encoders, while PromptSRC (Khattak et al., 2023b) enhances performance by knowledge distillation. Yao et al. (2024) incorporates class descriptions during tuning to improve discriminability. Lu et al. (2022) tunes text prompts using a classwise prompts pool, while Derakhshani et al. (2023) adds learnable noise to each text token, for generalization. These works mainly focus on standard image datasets with natural images and class semantics, overlooking realistic scenarios where the target data may exhibit substantial domain shifts or unfamiliar, specialized label semantics. Recent works have employed CLIP for cross-domain few-shot learning (CDFSL), which incorporates these challenges (Xiao et al., 2024; Brahma et al.). SRT (Xiao et al., 2024) trains the vision encoder with strong and weak augmentations, using multimodal mixup, while PromptMargin (Brahma et al.) uses a multimodal margin regularization to enforce uniform separation in the feature space. However, these methods take the episodic paradigm (N-way k-shot setting), and samples fixed number of classes in each episode, which is often unrealistic in real-world deployment. Moreover, since CLIP-based few-shot methods already adopt the k-shot setting ($k$ samples from all classes), it is unclear why CDFSL should be restricted to the episodic evaluation. *In this work, we explore few-shot adaptation of CLIP on the CDFSL benchmark (with severe domain and semantic shifts present) under the more practical k-shot all-class setting, bringing the evaluation closer to real-world deployment.*

**Stochastic neural networks.** Standard neural networks train weights deterministically as point-estimates. Contrarily, Bayesian Neural Networks (Neal, 2012; Blundell et al., 2015) model the weights as probability distributions, making them useful in handling uncertainty in predictions as well as learning robust representations. Stochastic classifiers have been explored in UDA (Lu et al., 2020), person re-identification (Yu et al., 2019), incremental learning (Kalla & Biswas, 2022) and DG (Zhou et al., 2023) in prior literature. *To the best of our knowledge, this is the first work which explores stochastic classifiers for few-shot adaptation of VLMs under extreme domain shifts.*

## 3 PROBLEM FORMULATION

Given few labeled training examples from a target dataset, the task is to adapt the CLIP model efficiently to this data. Formally, we have a support set $\mathcal{S} = \{(X_i, y_i)\}_{i=1}^{C \times k}$ from the target dataset

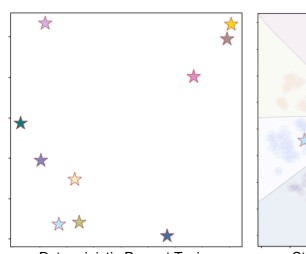 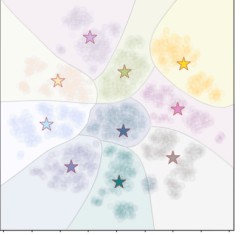

| Method | EuroSAT | ISIC |
|---|---|---|
| **1-Shot** | | |
| Deterministic Prompt Tuning | 73.30 | 27.50 |
| Stochastic Prompt Tuning $(\mu, \sigma^*)$ | 73.43 | 30.67 |
| Stochastic Prompt Tuning $(\mu^*, \sigma^*)$ | 67.40 | 22.67 |
| **8-Shots** | | |
| Deterministic Prompt Tuning | 86.80 | 46.53 |
| Stochastic Prompt Tuning $(\mu, \sigma^*)$ | 86.73 | 50.80 |
| Stochastic Prompt Tuning $(\mu^*, \sigma^*)$ | 88.03 | 50.93 |

Figure 1: Effect of deterministic vs stochastic prompt learning on the EuroSAT dataset with 1-shot per class. The projected class-specific image features are spread in fixed regions for stochastic learning, implicitly learning well-separated decision boundaries.

Figure 2: Stochastic prompt learning with two different sampling techniques on EuroSAT and ISIC datasets. The fixed mean approach $(\mu, \sigma^*)$ performs better in low shots, while sampling from a fully learnable distribution $(\mu^*, \sigma^*)$ performs better in higher shots.

containing $k$ samples from all the $C$ classes simultaneously. Here, $y_i \in \{0, 1\}^C$ is the corresponding ground truth label, and $k = \{1, 2, 4, 8, 16\}$, denotes the number of shots. The evaluation is done on the full test set. Here, in addition to the few-shot problem, the target dataset contains significant domain and label semantic shift from natural image datasets.

**Preliminaries** Here, we briefly describe CLIP classification and the base network used in this work. Let us denote the CLIP text and image encoders as $\mathcal{F}_t$ and $\mathcal{F}_v$ respectively. The input image $Xv \in \mathbb{R}^{C \times H \times W}$ is broken into patches $\{e_{CLS}, e_1, e_2, ..., e_M\}$ and fed to the image encoder to extract the image embedding $z_v = \mathcal{F}_v(X_v)$. Similarly, the text input (e.g. "A photo of a [CLS]") is tokenized as $X_t = \{t_{SOS}, t_1, t_2, ..., t_{CLS}, t_{EOS}\}$ and fed to the text encoder to get the text embedding $z_t = \mathcal{F}_t(X_t)$. During zero shot classification, the class text embeddings are matched with the image as follows: $\frac{exp(<z_v, z_t>/\tau)}{\sum_{i=1}^{C} exp(<z_v, z_{t_i}>/\tau)}$, where $C$ is the number of classes and $\tau$ is the temperature constant. The class with the highest similarity is output as the predicted class.

**Base Network of MIST:** We employ a multimodal prompt learning strategy, where learnable prompt vectors are appended to the image and textual branches. Specifically, let the set of learnable text and visual prompt vectors are denoted as $\theta_t = \{\theta_{t_1}, \theta_{t_2}, ..., \theta_{t_m}\}$ and $\theta_v = \{\theta_{v_1}, \theta_{v_2}, ..., \theta_{v_m}\}$ respectively, which are appended to the input text and image patches to form the modified inputs: $\tilde{X}_t = \{t_{SOS}, \theta_{t_1}, ..., \theta_{t_m}, t_1, t_2, ..., t_{CLS}, t_{EOS}\}$ and $\tilde{X}_v = \{e_{CLS}, \theta_{v_1}, ..., \theta_{v_m}, e_1, e_2, ..., e_M\}$. The feature embeddings from the CLIP encoders are now $\tilde{z}_t = \mathcal{F}_t(\tilde{X}_t)$ and $\tilde{z}_v = \mathcal{F}_v(\tilde{X}_v)$. Here, the trainable textual prompts are passed through a learnable projection layer $f_\phi$ to obtain the visual prompts, i.e., $\theta_v = f_\phi(\theta_t)$. Along with adding prompts to the inputs, we also adopt a deep prompting approach (Khattak et al., 2023a), where learnable prompt vectors are attached after every transformer block. When adapting to a downstream task, these multimodal prompts are trained in an end-to-end manner, keeping the CLIP encoders frozen.

## 4 THE PROPOSED FRAMEWORK

We aim to efficiently adapt pretrained CLIP using few samples from all classes under extreme domain and semantic shifts. To this end, we propose MIST (Fig. 4), which adds two novel modules: (i) Stochastic prompt learning to reduce overfitting, and (ii) Multiple prompts to relax the unimodal assumption of class distributions.

### 4.1 STOCHASTIC PROMPT LEARNING

Although existing prompt tuning methods optimize few parameters appended to inputs, limited training samples make large models like CLIP prone to overfitting (Khattak et al., 2023b). Moreover, with all target classes present, domain-specific or unfamiliar names can bring class features closer

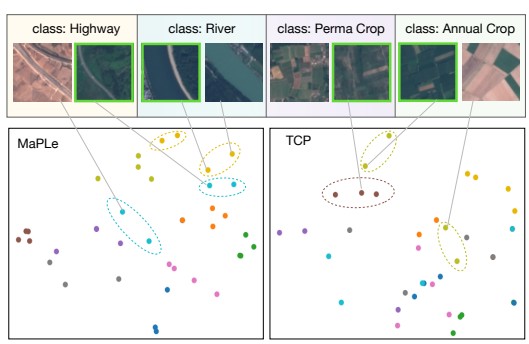

Figure 3: t-SNE visualization of image embeddings from MaPLe (Khattak et al., 2023a) (left) and TCP (Yao et al., 2024) (right) for the EuroSAT dataset. The classwise data distribution violates the unimodal assumption due to high interclass similarity and intraclass variations, and forms multiple disjoint clusters within the same class.

in representation space, reducing inter-class separability. We address this with a novel stochastic prompt learning strategy as described below.

**A Bayesian perspective.** A classifier network parameterized by $\theta$, can be viewed as a probabilistic model which models the conditional distribution $P(y|X, \theta)$, assigning a probability over class labels given an input image $X$. The parameters of the model can be estimated using maximum likelihood estimate (MLE) as follows:

$$\theta_{\text{MLE}} = \arg\max_\theta \log P(S \mid \theta) = \arg\max_\theta \sum_{i=1}^{k \times C} \log P(y_i \mid X_i, \theta) \tag{1}$$

which learns a point-estimate of the parameters, usually using gradient descent. However, when the training data is small (as in this case), MLE fails to generalize and overfits to the small number of observations. From a Bayesian perspective, the parameters can be estimated using a maximum a-posteriori (MAP) estimate, by introducing a prior over the parameters:

$$\theta_{\text{MAP}} = \arg\max_\theta \log P(\theta \mid S) = \arg\max_\theta \left[ \log P(S \mid \theta) + \log P(\theta) \right] \tag{2}$$

This provides an implicit regularization to the model, e.g., taking a Gaussian prior over $\theta$ is equivalent to $L_2$ regularization. The fully Bayesian approach helps estimate the posterior predictive distribution $P(y|X)$ for a given test sample as $P(y|X) = \mathbb{E}_{\theta \sim P(\theta|S)}[P(y|X, \theta)]$. This expectation effectively averages the predictions over an infinite number of classifiers. Since the true posterior $P(\theta|S)$ is intractible, variational methods (Hinton & Van Camp, 1993) have tried to estimate it with a parametric distribution $q(\theta|\psi)$, and learn $\psi$ via KL minimization, which is often computationally restrictive (Blundell et al., 2015). In contrast, inspired by Yu et al. (2019), we directly optimize the parameters $\psi$ using gradient descent on the final classification loss function, by sampling model weights from $q(\theta|\psi)$ at every epoch.

Specifically, we define $q(\theta|\psi)$ as a Gaussian distribution $\mathcal{N}(\mu, \sigma^2 I)$, and sample the text prompt weights as $\theta_t \sim \mathcal{N}(\mu, \sigma^2 I)$. After every iteration, we backpropagate the loss to the learnable parameters $\psi = \{\mu, \sigma\}$. Note that only the weights of the text prompt vectors are sampled, keeping rest of the model frozen. This helps to mitigate the uncertainty arising from scarce data, since each distinct sampled weight from this learnable distribution forms diverse decision boundaries for the few shot data, by allowing a richer exploration of the prompt parameter space. This provides an implicit regularization to the model without additional loss functions leading to more robust decision boundaries for the few-shot training samples.

**Reparameterization trick.** When $\theta_t$ is sampled from a Gaussian distribution, the loss cannot backpropagate to it, as the sampling process is non-differentiable. To avoid this, we use the reparameterization trick (Kingma et al., 2013), where we use a random sample from a standard Gaussian distribution $\epsilon \sim \mathcal{N}(0, I)$, and compute the prompt parameters as $\theta_t = \mu + \epsilon \cdot softplus(\sigma)$. This disentangles the randomness from the network and enables end-to-end training. The $softplus$ function $(softplus(\sigma) = log(1 + exp(\sigma)))$ ensures that the variance is non-negative.

We observe from Fig. 1 that in the deterministic case, passing the same examples through the trained model always projects them to the same points in the feature space. In contrast, the projections in the stochastic scenario spreads over a broader region, due to sampling of weights from the learned

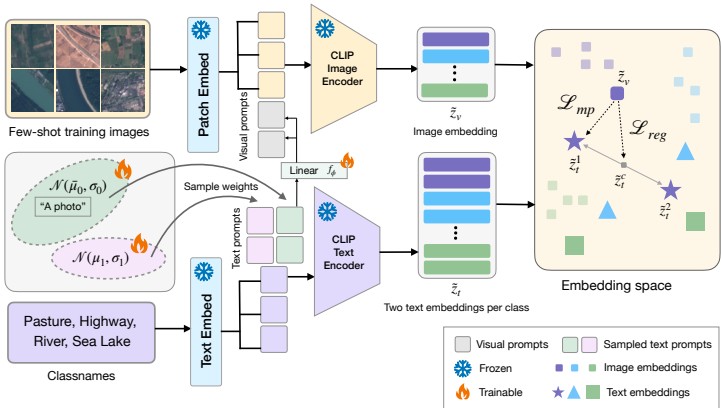

Figure 4: **Our proposed MIST framework.** We append two sets of prompts to the classnames, one sampled from a fixed mean $(\bar{\mu}_0, \sigma_0)$ and the other from a fully learnable Gaussian distribution $(\mu_1, \sigma_1)$. $f_\phi$ projects the text prompts to visual prompts. The loss term $\mathcal{L}_{mp}$ trains the distribution parameters $(\mu_1, \sigma_0, \sigma_1)$ and $f_\phi$ such that the image embedding is assigned to the closest text prototype of its respective class. The $\mathcal{L}_{reg}$ term prevents the two prompts from collapsing by enforcing diversity in the class-wise prompt training.

distribution. This variability implicitly encourages a margin between the class-specific features, resulting in more discriminative decision boundaries. Next, to verify this quantitatively, we consider two distinct strategies for sampling the text prompt parameters. First, we fix the mean of the Gaussian to the standard prompt "A photo", keeping the variance learnable. In the second case, both the mean and variance are learnable. The results on two datasets (Guo et al., 2020) are shown in Table 2. We observe that in low-shot setting, the first approach outperforms the second, while an opposite trend is seen in the higher-shot setting. This suggests that with very few training data (1-shot), directly optimizing the parameters of a distribution is challenging, but exploring variations around a standard prompt improves performance. On the contrary, learning the full distribution with more data outperforms the fixed mean approach. Hence, the two strategies complement each other in facilitating an efficient coverage of the prompt parameter space.

## 4.2 MIST: MULTIPLE STOCHASTIC PROMPT TUNING

**Limitations of existing methods.** Since CLIP is pretrained on web-scale data encompassing mainly natural images (Radford et al., 2021), the image encoder struggles to learn robust classwise features when faced with extreme domain and semantic shifts in the target dataset. In addition, intraclass diversity and interclass similarity in the images due to presence of many classes, results in disjoint clusters in visual features from the same class.

Consider an illustrative example of the EuroSAT dataset containing satellite images of various terrains. Here, distinct classes like "Highway" and "River" can have similar visual representatives in the few-shot training data, at the same time featuring diverse visuals from the same classes as shown in Fig. 3. Existing prompt tuning approaches represent each class with single learnable prompts, implicitly assuming that classwise visual features form single clusters. However, such a strategy is insufficient to represent the disjoint visual clusters, which may result from such challenging settings. To illustrate this, we consider two representative prompt-tuning methods, MaPLe (Khattak et al., 2023a) and TCP (Yao et al., 2024) and show their t-SNE visualizations in Fig. 3. We observe that the unimodal assumption is violated and the image embeddings of each class form multiple modes in the representation space.

**Our approach.** To address this, inspired from Allen et al. (2019); Afrasiyabi et al. (2021), we introduce a multiple prompt learning approach, where we represent each class with multiple prompt vectors. However, incorporating too many prompts for each class is not desirable since: (i) Each class contains few training samples, and many text embeddings per class could lead to overfitting on individual datapoints, resulting in loss of class level representations, and (ii) it introduces additional

learnable parameters and thus more computational overhead. As a balanced approach, we represent each class with two prompt vectors. Formally, let the embeddings corresponding to the two prompts for a particular class $CLS_k$ be denoted as $\tilde{z}_t^i$, where, $i = 1, 2$. Here, $\tilde{z}_t^i = \mathcal{F}_t(\tilde{X}_t^i)$, where, $\tilde{X}_t^i = \{t_{SOS}, \theta_t^i, t_1, ..., t_{CLS_k}, t_{EOS}\}$ is the $i^{th}$ text prompt for the class $CLS_k$. $\theta_t^i = \{\theta_{t1}^i, \theta_{t2}^i, ..., \theta_{tm}^i\}$ denotes the $i^{th}$ set of learnable prompt vectors for that particular class.

To simultaneously represent the underlying multimodal class distribution and mitigate overfitting, we stochastically model the parameters of the two prompts as described in Sec 4.1. Specifically, we incorporate the fixed mean, learnable variance approach on the parameters of the first prompt, simultaneously learning a full Gaussian distribution over the parameters of the second prompt:

$$\theta_t^1 \sim \mathcal{N}(\bar{\mu}_0, \sigma_0), \quad \theta_t^2 \sim \mathcal{N}(\mu_1, \sigma_1) \tag{3}$$

Here, $\bar{\mu}_0$ is a fixed vector corresponding to the text embedding of "A photo of a", and $\mu_1$, $\sigma_0$, $\sigma_1$ are learnable parameters. Now, we describe the training process of MIST.

**MIST Training:** For a particular image embedding $\tilde{z}_v$, we first find the closest text embedding of its respective class after every iteration, based on cosine similarity as follows:

$$i^* = \underset{i \in \{1,2\}}{\operatorname{argmax}} \ \ sim(\tilde{z}_t^i, \tilde{z}_v) \tag{4}$$

where, $sim(a, b) = \frac{a \cdot b}{\|a\|\|b\|}$ denotes the cosine similarity. During training, the image embedding is assigned to its closest prompt embedding by minimizing the following loss function:

$$\mathcal{L}_{mp} = -log \left( \frac{exp(sim(\tilde{z}_t^{i^*}, \tilde{z}_v))}{\sum_{j=1}^{2C} exp(sim(\tilde{z}_t^j, \tilde{z}_v))} \right) \tag{5}$$

where, $C$ is the number of classes, and $sim(\cdot)$ denotes the cosine similarity. To prevent the image embedding $\tilde{z}_v$ from being always assigned a single text prompt, and encourage diversity when training the two prompts within the same class, we minimize an additional regularization term to increase the cosine similarity of the image embedding to the centroid of the two text embeddings of its corresponding class:

$$\mathcal{L}_{reg} = -sim(\tilde{z}_v, \tilde{z}_t^c) \tag{6}$$

where, $\tilde{z}_t^c = \frac{1}{2}(\tilde{z}_t^1 + \tilde{z}_t^2)$ is the centroid of the prompt embeddings for the corresponding class and $sim(\cdot)$ represents the cosine similarity. Thus, the final objective function is:

$$\mathcal{L}_{total} = \mathcal{L}_{mp} + \mathcal{L}_{reg} \tag{7}$$

This loss function is used to optimize the Gaussian parameters $\mu_1$, $\sigma_0$ and $\sigma_1$, as well as the projection layers as:

$$\mu_1^*, \sigma_0^*, \sigma_1^*, \phi^* = \underset{\mu_1, \sigma_0, \sigma_1, \phi}{\operatorname{argmin}} \ \mathbb{E}_{(X,y) \sim D_{tgt}} \mathcal{L}_{total}(X, y) \tag{8}$$

**Inference:** After learning the parameters of the distribution, during inference, we can sample weights for the two text prompts as follows: $\theta_t^1 \sim \mathcal{N}(\bar{\mu}_0, \sigma_0^*)$ and $\theta_t^2 \sim \mathcal{N}(\mu_1^*, \sigma_1^*)$. For each class, we take the maximum logit among the two text prompts as the output prediction for that class.

## 5 EXPERIMENTAL RESULTS

Here, we extensively evaluate and compare the proposed method with state-of-the-art approaches.

**Datasets used:** For experiments we consider the BSCDFSL (Guo et al., 2020) benchmark, which is collected from real-world settings, and consists of four datasets: EuroSAT (Helber et al., 2019), ISIC (Codella et al., 2019), Plant Disease (Mohanty et al., 2016) and ChestX (Wang et al., 2017). These datasets cover a varying spectrum of domain shifts, along with specialized classnames, encompassing satellite, agricultural and medical images. For training, we consider few samples ($k = 1, 2, 4, 8, 16$) randomly selected from all the classes together and then evaluate the trained model on the full test set of all the datasets. The final accuracy is averaged over 3 random seeds.
**Implementation details:** We employ CLIP ViT-B/16 as the backbone similar to MaPLe (Khattak

| Method | EuroSAT | ISIC | PDisease | ChestX | Average |
|---|---|---|---|---|---|
| 1-shot | | | | | |
| CoOp (IJCV'22) | 51.87 | 22.77 | 24.73 | **22.83** | 30.55 |
| TaskRes (CVPR'23) | 64.67 | 19.70 | 36.57 | 10.97 | 32.98 |
| MaPLe (CVPR'23) | 73.30 | 27.50 | 51.53 | 14.60 | 41.73 |
| PromptSRC (ICCV'23) | 73.23 | 21.97 | **55.03** | 14.37 | 41.15 |
| CLAP (CVPR'24) | 61.46 | 26.61 | 47.22 | 15.94 | 37.81 |
| TCP (CVPR'24) | 64.30 | 27.80 | 49.37 | 14.93 | 39.10 |
| MIST (Ours) | **77.90** | **34.40** | 50.27 | 17.10 | **44.92** |
| 2-shot | | | | | |
| CoOp (IJCV'22) | 66.00 | 21.87 | 37.97 | 14.43 | 35.07 |
| TaskRes (CVPR'23) | 68.83 | 23.13 | 39.27 | 10.83 | 35.52 |
| MaPLe (CVPR'23) | 78.07 | 31.90 | 67.17 | 16.27 | 48.35 |
| PromptSRC (ICCV'23) | 79.53 | 29.47 | 68.07 | 12.70 | 47.44 |
| CLAP (CVPR'24) | 70.63 | 34.79 | 60.13 | **16.43** | 45.50 |
| TCP (CVPR'24) | 70.37 | **36.87** | 62.63 | 15.63 | 46.38 |
| MIST (Ours) | **81.57** | 36.37 | **69.60** | 13.90 | **50.36** |
| 4-shot | | | | | |
| CoOp (IJCV'22) | 66.53 | 25.00 | 42.67 | 17.93 | 38.03 |
| TaskRes (CVPR'23) | 72.40 | 21.40 | 39.35 | 10.27 | 35.86 |
| MaPLe (CVPR'23) | 84.03 | 37.17 | 77.07 | **19.73** | 54.50 |
| PromptSRC (ICCV'23) | 85.23 | 37.63 | 78.70 | 15.17 | 54.18 |
| CLAP (CVPR'24) | 76.43 | 34.37 | 65.11 | 18.98 | 48.72 |
| TCP (CVPR'24) | 76.77 | 37.37 | 67.97 | 17.07 | 49.80 |
| MIST (Ours) | **85.93** | **40.90** | **79.67** | 18.67 | **56.29** |
| 8-shot | | | | | |
| CoOp (IJCV'22) | 76.53 | 38.27 | 60.50 | 14.60 | 47.48 |
| TaskRes (CVPR'23) | 74.63 | 34.30 | 57.77 | 12.57 | 44.82 |
| MaPLe (CVPR'23) | 86.80 | 46.53 | 84.47 | 14.17 | 57.99 |
| PromptSRC (ICCV'23) | 88.37 | 42.47 | 86.80 | 14.93 | 58.14 |
| CLAP (CVPR'24) | 76.85 | 42.81 | 74.62 | 14.97 | 52.31 |
| TCP (CVPR'24) | 79.03 | 46.57 | 76.33 | 14.97 | 54.23 |
| MIST (Ours) | **88.63** | **52.70** | **87.47** | **16.50** | **61.33** |
| 16-shot | | | | | |
| CoOp (IJCV'22) | 82.83 | 43.40 | 69.90 | **18.80** | 53.73 |
| TaskRes (CVPR'23) | 79.90 | 38.10 | 69.40 | 12.87 | 50.07 |
| MaPLe (CVPR'23) | 92.80 | 55.53 | 89.93 | 13.90 | 63.04 |
| PromptSRC (ICCV'23) | 92.55 | 55.17 | 91.40 | 14.83 | 63.49 |
| CLAP (CVPR'24) | 82.96 | 49.43 | 78.27 | 17.47 | 57.03 |
| TCP (CVPR'24) | 84.93 | 52.83 | 80.63 | 16.53 | 58.73 |
| MIST (Ours) | **93.57** | **60.30** | **91.73** | 14.77 | **65.09** |

Table 1: Performance comparison (average accuracy (%) over 3 seeds) of the proposed MIST with the state-of-the-art approaches for $k = 1, 2, 4, 8, 16$ shots from each class.

et al., 2023a). The learnable context length of the text and vision inputs are set as 2, and deep prompts are incorporated upto a depth of 9. The $f_\phi \in \mathbb{R}^{512 \times 768}$ is a single linear layer projecting text to visual prompts. The model is trained using SGD optimizer for 150 epochs with a learning rate of 0.0035 and a batch size of 4. All experiments are conducted on a NVIDIA RTX A5000 GPU.

## 5.1 COMPARISON WITH STATE-OF-THE-ART METHODS

To validate the effectiveness of our approach, we compare our proposed MIST with several recent CLIP-based efficient transfer learning methods for varying number of shots. Specifically, we compare with 1) **CoOp** (Zhou et al., 2022) and **TCP** (Yao et al., 2024), which employ prompt tuning on the text branch; 2) **MaPLe** (Khattak et al., 2023a) and **PromptSRC** (Khattak et al., 2023b) which utilize a multimodal prompt tuning approach; 3) **TaskRes** (Yu et al., 2023) where task-specific adapters are tuned keeping the base text classifier fixed; 4) **CLAP** (Silva-Rodriguez et al., 2024) uses a linear-probing approach and mainly addresses the absence of validation sets in FSL.

For fair comparison, we run all the methods (using the official, publicly available codes) on the ViT-B/16 backbone and report the results in Table 1. We list some observations below:

(i) Among the competing methods, MaPLe and PromptSRC achieves the highest performance on average, closely followed by TCP. Their improved performance suggests the effectiveness of multimodal prompt tuning in handling distribution shifts over text prompt tuning, which was also observed by Khattak et al. (2023a). This observation is further supported by our results;

(ii) Although, CLAP is a recent approach, it mainly focuses on the validation problem of FSL. The reduced performance of CLAP highlights the limitations of linear probing, which does not utilize the text information for handling significant semantic and domain shifts;

(iii) As the number of shots increases, the multimodal prompt tuning approaches like MaPLe, PromptSRC and MIST outperforms other methods by larger margins, suggesting that training more

| Dataset | MaPLe | PrSRC | TCP | MIST (Ours) |
|---------|-------|-------|-----|-------------|
| EuroSAT | 72.90 | 74.10 | 57.80 | **76.90** |
| ISIC | 14.80 | 16.10 | 13.13 | **16.73** |

Table 3: Performance comparison (%) of MIST with state-of-the-art methods on the class-imbalanced setting, with varying data samples from each class.

| | | EuroSAT | ISIC |
|--------|-----------|---------|------|
| 1-shot | MaPLe | 73.30 ±3.84 | 27.50 ±10.19 |
| | PromptSRC | 73.23 ±3.75 | 21.97 ±6.09 |
| | TCP | 64.30 ±3.24 | 27.80 ±6.55 |
| | MIST (Ours) | 77.90 ±**2.63** | 34.40 ±**3.56** |
| 2-shots | MaPLe | 78.07 ±5.87 | 31.90 ±6.08 |
| | PromptSRC | 79.53 ±2.76 | 29.47 ±6.50 |
| | TCP | 70.37 ±2.31 | 36.87 ±8.29 |
| | MIST (Ours) | 81.57 ±**1.84** | 36.37 ±**3.96** |

Table 4: MIST exhibits significantly lower variance across three different random seeds compared to other approaches.

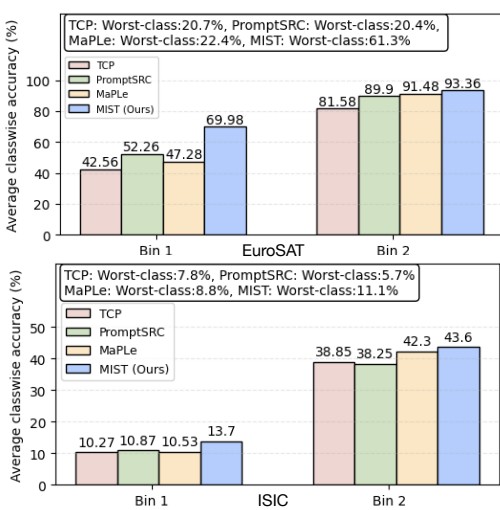

Figure 5: Generalization to classes: The class-wise accuracies are sorted and divided into 2 bins. MIST outperforms the other methods in both bins and increases the worst-class accuracy for the same seed in the challenging 1-shot setting.

parameters is more effective for higher shots.

(iv) Although all the methods show similar performance on the ChestX dataset, their overall accuracies are extremely low due to the large domain shift. However, text prompt tuning methods perform slightly better than the multimodal counterparts. This maybe because ChestX contains greyscale images, where additional prompt tuning in the vision branch degrades the performance.

Overall, our proposed MIST framework outperforms the other methods significantly, giving consistent average gains of 3.19%, 2.01%, 1.79%, 3.19%, 1.60% on $k = 1, 2, 4, 8, 16$ shots respectively, over the best performing methods. The significant improvement for the 1-shot case highlights the effectiveness of our approach in mitigating overfitting in extremely low data scenarios.

| Method | EuroSAT | ISIC |
|--------|---------|------|
| Base Network | 73.30 | 27.50 |
| + Stochastic Prompt Learning ($\mu, \sigma^*$) | 73.43 | 30.67 |
| + Multiple Stochastic Prompting | 74.27 | 27.00 |
| + Multiple Stochastic Prompting + $\mathcal{L}_{reg}$ (MIST) | **77.90** | **34.40** |

Table 2: Ablation study (1-shot): All the proposed components collectively enhance the overall performance.

## 5.2 ADDITIONAL ANALYSIS

Here we perform additional analysis and ablation studies to further validate our proposed framework. For the analysis, we compare with MaPLe, PromptSRC since they use multimodal prompts and are better suited for this task and TCP, since it is the state-of-the-art prompt tuning approach.

**1) Class-imbalanced learning:** Here, we explore an even more challenging scenario, where the number of labeled examples may vary across classes, reflecting real-world datasets. The standard few-shot settings in literature assume an idealistic scenario where each class has exactly $k$ training samples, overlooking the effect of class imbalance. To create such a setting, we perform data sampling in a cyclic manner, e.g., we take $\{1, 2, 4, 8, 1, 2, ...\}$ from each class of the target dataset for training. The model is then evaluated on the entire test set. The results on two representative datasets, EuroSAT and ISIC in Table 3 shows that the proposed MIST outperforms the other methods even under class-imbalanced conditions, highlighting its effectiveness.

**2) Sensitivity to training samples:** Model performance depends on the few training samples available. A robust model should exhibit low variance across sampling strategies. We average results

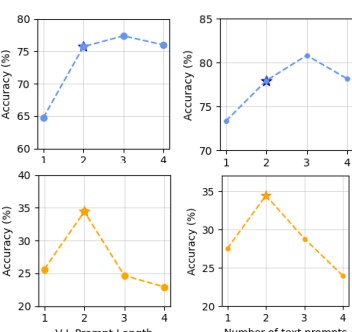

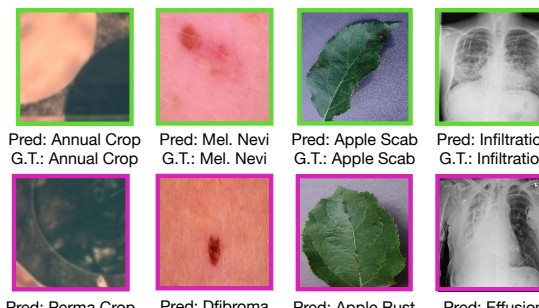

Figure 6: Effect of prompt length (left) and number of text prompts per class (right) for EuroSAT (blue) and ISIC (orange) (1-shot).

Figure 7: Qualitative results: From left to right, shows predictions on EuroSAT, ISIC, Plant Disease and ChestX respectively. Green denotes correct while red denotes incorrect predictions.

over 3 random seeds and report variance for EuroSAT and ISIC in Table 4. MIST achieves higher accuracy with significantly lower variance, demonstrating robustness to sample variation.

**3) Generalization to all classes:** In practical scenarios, the overall accuracy is often not a reliable metric to understand the model's ability to represent difficult classes. Here, we study the effectiveness of our approach in learning generalized class boundaries and modeling all the complex class distributions. Specifically, we first sort the class-wise accuracies in ascending order. The classes are then divided into two bins in this order to highlight the accuracy gain in both the lower as well as the higher bin. The comparison with the other methods are shown in Figure 5 for one random seed (same for all methods). We observe that the proposed MIST improves the accuracies in both the bins, while also increasing the worst-class accuracy, which indicates that MIST learns more generalized class representations.

**4) Ablation Study:** Table 2 analyzes various components of MIST. Starting from the base method MaPLe, adding stochasticity via a learnable Gaussian prompt (fixed mean) reduces overfitting and improves performance. Adding a second learnable prompt without $\mathcal{L}reg$ boosts EuroSAT but lowers ISIC performance, likely due to classifier collapse (Afrasiyabi et al., 2021; Tian et al., 2024). Including $\mathcal{L}reg$ consistently improves results across both datasets.

**5) Number of text prompts & Prompt Length:** MIST utilizes two prompts per class sampled from learnable Gaussian distributions. Here we study the effect of adding more text prompts. We keep one prompt with a fixed mean, and the others sampled from fully learnable distributions. Figure 6 (right) shows that the performance starts decreasing after two (ISIC) or three prompts (EuroSAT), suggesting overfitting from the increasing number of learnable parameters. Further, addition of more classifiers introduces increased computational overhead and longer training time. Similarly increasing the number of learnable prompt vectors also degrades accuracy (Figure 6 (left)). We used 2 learnable prompts for all experiments.

**Qualitative Results:** We illustrate the inherent challenges of these datasets in Figure 7, along with some of the predictions from the proposed MIST framework.

**Limitations.** While MIST outperforms state-of-the-art methods across all datasets, its performance slightly drops on the grayscale ChestX dataset, likely because of the additional visual prompts. In such cases, methods relying solely on textual prompts may prove more effective.

## 6 CONCLUSION

In this work, we propose a novel framework, MIST for adapting foundation VLMs like CLIP to realistic few-shot scenarios characterized by extreme domain and label semantic shifts. Motivated by the limitations of existing parameter efficient fine-tuning approaches, we incorporate multiple text prompts per class, modeled by distinct learnable Gaussian distributions to represent the inherent multimodal class distributions as well as mitigate overfitting. Extensive experiments on multiple benchmarks as well as additional analysis show the effectiveness of our proposed approach compared to state-of-the-art methods.

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

# A   APPENDIX

## A.1   MORE DATASET DETAILS

For all the experiments, we have considered the four datasets from the BSCDFSL (Guo et al., 2020) benchmark. This benchmark consists of images and classnames from various specialized domains like medical, satellite and agricultural fields, and exhibits extreme domain and label semantic shifts from natural image datasets. We provide more details on each of these datasets below:

1) **EuroSAT (Helber et al., 2019):** This dataset consists of satellite images of various terrains and comprises of 10 classes, namely, Annual Crop, Forest, Herbaceous Vegetation, Permanent Crop, Residential Buildings, Pasture, Industrial buildings, Highway, River, Sea-Lake.

2) **ISIC (Codella et al., 2019):** Consists of dermascopic skin disease images and has 7 classes, namely, Melanoma, Melanocytic Nevi, Basal Cell Carcinoma, Actinic Keratosis, Benign Keratosis, Dermatofibroma, Vascular Lesions.

3) **Plant Disease (Mohanty et al., 2016):** Contains images of leaf diseases across 38 classes, e.g., Apple Scab, Apple Black Rot, Apple Cedar Rust, Apple Healthy, Blueberry healthy, Cherry Powdery Mildew, Cherry Healthy, and so on.

4) **ChestX (Wang et al., 2017):** This dataset comprises of greyscale chest X-Ray images across 7 classes, namely, Atelectasis, Cardiomegaly, Effusion, Infiltration, Mass, Nodule, Pneumothorax.

The datasets ranked in decreasing order of similarity with ImageNet (Deng et al., 2009) are as follows: Plant Disease > EuroSAT > ISIC > ChestX.

## A.2   EFFECT OF PROMPT SAMPLING STRATEGIES

To enable end-to-end optimization of the prompt weights, we employ the standard reparameterization trick, which allows backpropagation through the non-differentiable sampling process. Specifically, we sample prompt weights from a learnable Gaussian distribution, and parameterize the variance using the *softplus* function to ensure stability during training. We choose Gaussian sampling due to its simplicity and ease of reparameterization, which has also been successfully explored in prior works like Universal Domain Adaptation (Lu et al., 2020) and Domain Generalization (Zhou et al., 2023). In contrast, alternative distributions such as Gaussian Mixture Models (GMMs) or Laplace distribution introduce additional complexity and instability during training. We report a comparison of these different sampling strategies in Table 5, and observe that, overall the Gaussian sampling consistently outperforms the other strategies in our stochastic prompt learning framework.

| Prompt Sampling | EuroSAT | ISIC |
|---|---|---|
| Gaussian (Ours) | 77.90 | 34.40 |
| GMM (N=4) | 78.57 | 22.10 |
| Laplace | 11.10 | 11.30 |

Table 5: Effect of different sampling strategies for prompt weights.

| | MaPLe | PromptSRC | TCP | MIST (Ours) |
|---|---|---|---|---|
| Training | 1172 | 2168 | 778 | 1348 |
| Inference | 1608 | 2994 | 4946 | 1610 |

Table 6: GPU memory requirements (MB) for training and inference.

## A.3   MEMORY CONSUMPTION

We report the comparative GPU memory consumptions for the 1-shot case in Table 6. Here, we observe that our proposed MIST utilizes slightly more memory than MaPLe (Khattak et al., 2023a) during training, due to addition of learnable parameters, but utilizes the same memory during inference. However, PromptSRC (Khattak et al., 2023b) takes up much more memory during both

training and inference. TCP (Yao et al., 2024) takes slightly less memory due to fewer learnable parameters during training, but consumes a huge memory during inference, due to loading of the frozen CLIP model. Overall, our proposed MIST framework fairs comparably to MaPLe, and is more suitable for real-world deployment compared to the other methods.

**Use of Large Language Models (LLMs):** Certain parts of the manuscript, including sentence polishing, paraphrasing, and clarity improvements, were assisted using ChatGPT by OpenAI. All scientific content, ideas, and results remain the authors' own.

