# OpenReview forum: "MIST: Multiple Stochastic Prompt Tuning for Few-shot Adaptation under Extreme Domain Shift"
_ICLR.cc/2026/Conference — Submitted to ICLR 2026_

### Official Review · Reviewer_QjNq · 2025-10-30

**Soundness:** 2
**Presentation:** 3
**Contribution:** 2
**Rating:** 4
**Confidence:** 4

**Summary:**

The paper proposes a prompt tuning method for cross-domain few-shot adaptation using CLIP. Existing approaches face two key limitations: (1) they often overlook domain shifts that involve both visual appearance and label semantics, and (2) their evaluations typically exclude part of the target classes. To address these issues, the authors introduce two key techniques. First, they model prompt parameters stochastically using Gaussian distributions rather than deterministic vectors, improving robustness to domain and semantic variation. Second, they employ multiple prompts per class to better capture the diverse semantic representations within each category. The proposed method is extensively evaluated across multiple standard benchmarks

**Strengths:**

- The paper is well-written, allowing readers to easily follow the logical flow and understand the motivation as well as the proposed method at both the conceptual and technical levels. For example, in **Section 4.2**, the authors clearly introduce the limitation that motivates their approach, then explain the conceptual idea (“why” and “what”) before presenting the detailed technical design (“how”). In addition, **Figure 4** effectively visualizes the overall pipeline of the proposed framework.

- The experimental evaluation is comprehensive, including comparisons with state-of-the-art methods, detailed ablation studies, and hyperparameter analyses such as class imbalance, number of prompts, and prompt length.

**Weaknesses:**

- The problem setting in the Introduction is not clearly defined. Adapting CLIP to downstream target datasets can correspond to several existing problem formulations (e.g., few-shot adaptation, domain generalization, or open-world transfer). For instance, models such as Few-Shot Test-Time Domain Adaptation (FSTT-DA)[1] address few-shot adaptation with unlabeled target data under challenging domain and semantic shifts [2]. In this paper, the stated task is cross-domain few-shot learning, where both the domains and classes differ, but this distinction is not explicitly articulated. The authors are encouraged to include one or two sentences clarifying the setting in terms of domain, class, and label availability, and to highlight how it differs from related formulations. In addition, in Line 41, the definition of cross-domain few-shot learning may confuse readers unfamiliar with the meta-learning paradigm. It would help to explain K-way-N-shot evaluation in simple terms.

- The conceptual idea of using multiple learnable prompt embeddings to represent knowledge for a specific domain or class is not entirely novel. For example, prior works [3] have introduced prompt pools containing multiple learnable prompts whose weighted combination captures domain-specific knowledge. The same concept can be directly extended to class-specific representations. The authors should clarify these conceptual overlaps in the Related Work section and explicitly emphasize how their method differs or improves upon these earlier approaches.

- The proposed method is evaluated only with CLIP ViT-B/16, leaving uncertainty about whether the improvements generalize to other backbones. While computational constraints are understandable, testing on at least one additional architecture—such as ViT-L/14—on a representative benchmark would strengthen the claim of generality.

[1] Meta-dmoe: Adapting to domain shift by meta-distillation from mixture-of-experts. NeurIPS 2022

[2] WILDS: A Benchmark of in-the-Wild Distribution Shifts. ICML 2021

[3] Adapting to Distribution Shift by Visual Domain Prompt Generation. ICLR 2024

**Questions:**

- Could the authors explicitly clarify the problem setting of this work? How are domain, class, and label availability defined in this setting?

- How does this problem formulation differ from existing setups such as FSTT-DA or other few-shot adaptation methods?

- In Line 41, the definition of cross-domain few-shot learning may be unclear to readers without a meta-learning background. Could the authors briefly explain K-way-N-shot evaluation in simpler terms for clarity?

- Did the authors compare against or draw inspiration from existing methods that use multiple prompts for domain or class-specific adaptation? If so, please clarify these differences in Related Work or Ablation Studies.

- The experiments are conducted only on CLIP ViT-B/16. Can the authors comment on whether similar performance trends are expected on larger or different architectures such as ViT-L/14?

---

> ### Author Response · Authors · 2025-12-02
> **Response to Reviewer QjNq (1/2)**
>
> We thank the reviewer for the insightful comments. We are encouraged that the reviewer finds the paper well-written with a clear logical flow of the motivation leading to the method, and that the experimental evaluation is comprehensive. We address the reviewer concerns below:
>
> >**The problem setting in the Introduction is not clearly defined. Adapting CLIP to downstream target datasets can correspond to several existing problem formulations (e.g., few-shot adaptation, domain generalization, or open-world transfer). For instance, models such as Few-Shot Test-Time Domain Adaptation (FSTT-DA)[1] address few-shot adaptation with unlabeled target data under challenging domain and semantic shifts [2]. In this paper, the stated task is cross-domain few-shot learning, where both the domains and classes differ, but this distinction is not explicitly articulated. The authors are encouraged to include one or two sentences clarifying the setting in terms of domain, class, and label availability, and to highlight how it differs from related formulations. In addition, in Line 41, the definition of cross-domain few-shot learning may confuse readers unfamiliar with the meta-learning paradigm. It would help to explain K-way-N-shot evaluation in simple terms.**
>
> We clarify the problem setting below. Our work focuses on few-shot adaptation of CLIP in the presence of significant domain and label semantic shifts, i.e., the target dataset images have significantly different images and the classnames belong to specialized domains not seen during pretraining (e.g., ISIC dataset). In our setting, the model is adapted on k labeled samples from each class and evaluated on the full test set of the target dataset. E.g., the ISIC dataset has 7 classes, so we train on k-samples (k can be 1,2,4,8,16) from all these classes (total $7\times k examples), and then evaluated on the full test set of ISIC. This setup is closely related to the CLIP-based parameter-efficient finetuning works, but they mainly focus on natural, standard image benchmarks, leaving extreme real-world domain shifts largely unexplored. While some recent CLIP-based works have explored the domain-shift regime (usually called cross-domain few-shot learning), they mainly perform meta-testing on a large number of sampled episodes. Each episode is constructed by sampling fixed N classes and k images from each class (N-way k-shot setting), and the model is adapted on these Nk samples, followed by evaluation on a few query samples per episode. Importantly, unlike us, these works do not consider all the classes available in the target dataset. These distinctions differentiate our setting from prior works.
>
> Regarding the cited paper, the referenced method operates under a test-time or unsupervised adaptation paradigm, where the model is adapted using unlabeled target data within a meta-learning framework. This formulation is conceptually different from our supervised few-shot adaptation setting, where labeled examples are available for each target class. For ease of understanding, we show a Table below with a summarization of the distinctions between the different settings.
>
> | Setting                   | Source-domain training | Target train data        | Target test data    | Labels in target train? | Test-time unlabeled adaptation | Episodic? | All classes at once? |
> |---------------------------|------------------------|---------------------------|----------------------|--------------------------|-------------------------------|-----------|-----------------------|
> | **Ours**                  | No                     | k-shot per class          | Full test set        | Yes                      | No                            | No        | Yes                   |
> | **FSTT-DA**               | Yes                    | None                      | Few unlabeled target samples (for adaptation) | No | Yes (unlabeled)         | Yes       | No (episodic)    |
> | **CDFSL**  | Yes (meta-training)    | k-shot per episode (sampled classes) | Few query samples per episode | Yes | No | Yes | No (episodic) |
> | **Domain Generalization** | Yes                    | None                      | Full target test     | No                       | No                            | No        | Yes                   |

---

> ### Author Response · Authors · 2025-12-02
> **Response to Reviewer QjNq (2/2)**
>
> >**The conceptual idea of using multiple learnable prompt embeddings to represent knowledge for a specific domain or class is not entirely novel. For example, prior works [3] have introduced prompt pools containing multiple learnable prompts whose weighted combination captures domain-specific knowledge. The same concept can be directly extended to class-specific representations. The authors should clarify these conceptual overlaps in the Related Work section and explicitly emphasize how their method differs or improves upon these earlier approaches.**
>
> We clarify the conceptual distinction between our work and prior “prompt pool” approaches.
> The term prompt pool has been used in different contexts in prior literature, often in varying ways and purposes:
> (a) Zero-shot classification:
> Methods like [1] use multiple descriptive text prompts generated by LLMs and average or ensemble them to obtain a single prompt representation for each class.
> (b) Continual learning:
> In methods like [2], a large shared prompt pool is maintained, where each prompt is designated with a key. During training and inference, a relevant subset of prompts is retrieved and concatenated, which are then prepended to the visual patch tokens. Here, the multiple prompts are dynamically selected based on the input image, but they still function collectively as one prompt for the input and are not class-specific.
> (c) FSTT-DA [3] (cited by the reviewer): This method utilizes the unlabeled test samples from the target domain to generate visual prompts using a conditional prompt generator, and merges them into a single domain-specific prompt. This prompt is used to encode the domain-specific characteristics.
>
> In all these cases, although multiple prompts appear in various ways, they are not used as multiple independent class-specific prompt embeddings. Instead, they are eventually aggregated or combined into a single effective prompt representation for the model.
>
> Our approach is fundamentally different:
> We explicitly learn multiple prompts per class to capture multiple peaks in the visual feature distribution caused by extreme domain shifts. These prompts remain distinct and is used to directly model the class-specific image representations. Furthermore, we model each prompt as a learnable Gaussian distribution, to mitigate overfitting, not explored in prior prompt pool methods.
>
> [1] Pratt et. al., "What does a platypus look like? Generating customized prompts for zero-shot image classification", in ICCV 2023.
>
> [2] Wang et. al., "Learning to Prompt for Continual Learning", in CVPR 2022.
>
> [3] Chi et. al., "Adapting to Distribution Shift by Visual Domain Prompt Generation", ICLR 2024.
>
> >**The proposed method is evaluated only with CLIP ViT-B/16, leaving uncertainty about whether the improvements generalize to other backbones. While computational constraints are understandable, testing on at least one additional architecture—such as ViT-L/14—on a representative benchmark would strengthen the claim of generality.**
>
> Prior methods for parameter-efficient prompt-tuning of CLIP primarily report results using the ViT-B/16 backbone, since prompt-tuning behavior is known to vary across architectures due to differences such as visual and text feature dimensionality. However, following the reviewer’s suggestion, we additionally evaluate our method on the CLIP ViT-L/14 backbone and report the results below (Table A). We observe that MIST generalizes well to this larger backbone and continues to outperform existing prompt-tuning baselines.
>
> | Setting | Method | Eurosat | ISIC  |
> |---------|--------|---------|-------|
> | 1-shot | TCP   | 70.97   | 30.83 |
> |         | PromptSRC  | 76.43   | 22.23 |
> |         | MIST  | 76.60   | 29.27 |
> ||||
> | 2-shots | TCP   | 75.80   | 36.83 |
> |         | PromptSRC  | 79.93   | 33.70 |
> |         | MIST  | 80.63   | 37.73 |
>
> Table A: Performance comparison on the ViT-L/14 backbone.

---

### Official Review · Reviewer_p6n6 · 2025-11-01

**Soundness:** 3
**Presentation:** 2
**Contribution:** 2
**Rating:** 4
**Confidence:** 4

**Summary:**

This paper proposes MIST (Multiple Stochastic Prompt Tuning) to address few-shot adaptation of Vision-Language Models (VLMs) like CLIP under domain shifts and label semantic misalignment. MIST introduces multiple learnable prompt vectors per class modeled as Gaussian distributions to capture multimodal visual features, enhancing generalization and reducing overfitting in data-scarce scenarios. Experiments demonstrate significant improvements over SOTA methods, particularly in ultra-low-data regimes (1-shot).

**Strengths:**

1. MIST diagnoses fragmented feature representations under extreme domain shifts and proposes multiple Gaussian-distributed prompts per class to form diverse decision boundaries—a principled approach beyond standard prompt tuning.
2.  Significant improvements over methods across multiple benchmarks, especially in 1-shot scenarios, with demonstrated robustness to class imbalance and good generalization.

**Weaknesses:**

1. Weak Motivation and Limited Novelty: The method lacks clear motivation for why Gaussian-distributed multiple prompts address extreme domain shifts. The approach resembles existing GMM-based strategies without sufficient differentiation. Why Gaussian over other probabilistic models?

2. Incomplete Experiments: Baselines are outdated; missing recent few-shot adaptation methods. Limited to four benchmarks (EuroSAT, ISIC, Plant Disease, ChestX). No evaluation on standard OOD benchmarks (ImageNet-R, ImageNet-A, CIFAR-10-C, DomainNet, VisDA, etc.). Missing cross-dataset generalization experiments.

**Questions:**

Have you evaluated on other OOD benchmarks? How does MIST compare to recent domain generalization and OOD adaptation methods?

---

> ### Author Response · Authors · 2025-12-02
> **Response to Reviewer p6n6 (1/2)**
>
> We appreciate the reviewer's acknowledgement that the proposed method is a principled approach beyond standard learning, and that we show significant improvement over multiple benchmarks , with good generalization.
>
> We address the reviewer concerns below:
>
> >**The method lacks clear motivation for why Gaussian-distributed multiple prompts address extreme domain shifts. The approach resembles existing GMM-based strategies without sufficient differentiation. Why Gaussian over other probabilistic models?**
>
> We appreciate the reviewer’s question regarding the motivation for using Gaussian-distributed multiple prompts. We clarify our design choices below.
>
> 1) Why Gaussian distributed Multiple prompts: Under strong, distribution shifts in visual appearances, the visual features form fragmented well separated clusters instead of a single class-specific cluster in the representation space. A single text prompt per class (which acts as a classifier) is often insufficient to capture these diverse representations and hence motivates the need of multiple class-specific prompts. Further, we model these prompts as distinct parametric Gaussian models to mitigate overfitting due to few-shot adaptation as well as model uncertainty. The motivation and limitations leading to our proposed approach is detailed in Sec. 4.1 and 4.2.
>
> 2) Distinction with GMM-based methods: To the best of our knowledge, there are no prior works on CLIP adaptation using GMM-based modeling. While unsupervised learning methods may model the visual features using GMMs or clusters, our method models learnable prompt embeddings as Gaussians, and shifts the probabilistic modeling from the data space to the classifier space in a supervised learning setup. This design is fundamentally different and well suited for parameter-efficient CLIP adaptation as we demonstrate through our experiments.
>
> 3) Why Gaussian over other probabilistic models: We primarily use Gaussian modeling since it allows easier reparameterization-based sampling, making optimization stable and efficient.
> Alternate distributions (e.g., mixture densities) introduce unnecessary complexity and are unstable with few-shot data as we show below (Table A).
>
> | Sampling Method | EuroSAT | ISIC  |
> |-----------------|---------|-------|
> | Gaussian (Ours) | 77.90   | 34.40 |
> | GMM (components=4)       | 78.57   | 22.10 |
> | Laplace         | 11.10   | 11.30 |
>
> Table A: Effect of sampling techniques for the multiple prompts in MIST.

---

> ### Author Response · Authors · 2025-12-02
> **Response to Reviewer p6n6 (2/2)**
>
> >**Incomplete Experiments: Baselines are outdated; missing recent few-shot adaptation methods. Limited to four benchmarks (EuroSAT, ISIC, Plant Disease, ChestX). No evaluation on standard OOD benchmarks (ImageNet-R, ImageNet-A, CIFAR-10-C, DomainNet, VisDA, etc.). Missing cross-dataset generalization experiments.**
>
> We have provided comparisons with more recent methods in response to Reviewer Pyp4. For reference, we also report it below in Table A.
>
> | Shots     | Method       | EuroSAT | ISIC  | PlantDisease | ChestX | Average |
> |-----------|--------------|---------|-------|--------------|--------|---------|
> | 1-shot    | 2SFS (CVPR 25)        | 73.7    | 26.23 | 43.9         | 13.43  | 39.32  |
> |   | PromptMargin (TMLR 25)  | 68.10 | 30.37 | 49.83 | 15.00    | 40.83 |
> |           | MIST         | 77.9    | 34.4  | 50.27        | 17.1   | 44.92 |
> |           |              |         |       |              |        |         |
> | 2-shots   | 2SFS (CVPR 25)         | 80.44   | 32.67 | 57.97        | 14.74  | 46.46  |
> |  | PromptMargin (TMLR 25)  | 75.83   | 35.53 | 60.03        | 15.33  | 46.68 |
> |           | MIST         | 81.57   | 36.37 | 69.6         | 13.9   | 50.36   |
> |           |              |         |       |              |        |         |
> | 4-shots   | 2SFS (CVPR 25)         | 85.5    | 36.62 | 68.96        | 19.05  | 52.53 |
> |  | PromptMargin (TMLR 25)  | 83.50    | 36.67 | 62.33        | 16.83  | 49.83 |
> |           | MIST         | 85.93   | 40.9  | 79.67        | 18.67  | 56.29 |
>
> Table A: Comparison with more recent baselines.
>
> The most relevant and established benchmark for evaluating few-shot adaptation under extreme domain shift is the BS-CDFSL benchmark, which we have evaluate on: EuroSAT, ISIC, PlantDisease, and ChestX. These datasets come from real-world specialized domains (remote sensing, medical imaging, agriculture), exhibit large domain shifts from natural images, and involve label semantic shifts, since class names are domain-specific and not seen in CLIP pretraining.
>
> The datasets mentioned by the reviewer are widely used for domain generalization (DG) tasks and not few-shot adaptation; here we train the model on a source domain data and evaluate in a zero-shot manner on shifted domains. Adapting them for few-shot evaluation requires non-trivial design choices, such as: (a) choosing which domains serve as “train” vs. “test”, and (b) constructing class-balanced few-shot splits. Thus, these datasets are not standard for few-shot adaptation and are fundamentally mismatched with our problem formulation.
>
> However, to address the reviewer’s concerns, we adapted a subset of these datasets into a few-shot setting where possible and report the improvement of our added components over the Maple baseline (Table B). This demonstrates that our method continues to generalize even when evaluated in a non-standard setting.
>
> | Shots  | Method                 | VisDA | CIFAR10-C | FGVCAircraft |
> |--------|-------------------------|-------|-----------|---------------|
> | 1-shot | Maple                   | 78.57 | 64.90     | 22.87         |
> |        | MIST               | 82.55 | 65.03     | 28.03         |
> |        |                         |       |           |               |
> | 2-shot | Maple                   | 80.33 | 67.13     | 30.37         |
> |        | MIST         | 81.00 | 66.57     | 31.63         |
>
> Table B: Other datasets (not specific to few-shot learning).
>
> We also report the cross-dataset generalization performance (Table C), where we first train our model on CIFAR-10 (1-shot) and then transfer this trained model to the domain-shifted datasets in a zero-shot manner.
>
> | Method | EuroSAT | ChestX |
> |--------|---------|--------|
> | PromptSRC   | 42.63   | 15.00 |
> | TCP    | 44.60   | 11.13  |
> | MIST   | 52.97   | 14.50  |
>
> Table C: Cross dataset generalization.

---

### Official Review · Reviewer_Pyp4 · 2025-11-01

**Soundness:** 2
**Presentation:** 3
**Contribution:** 2
**Rating:** 2
**Confidence:** 3

**Summary:**

This paper proposes MIST (Multiple Stochastic Prompt Tuning), a method for adapting CLIP-like vision-language models to few-shot tasks under extreme domain and label semantic shifts. MIST introduces (i) stochastic prompt learning (each prompt represented as a Gaussian distribution) to improve generalization and (ii) multiple prompts per class to model multimodal feature distributions. The method is evaluated on the BSCDFSL benchmark (EuroSAT, ISIC, PlantDisease, ChestX) and shows improved performance on most datasets over SOTA methods published up to 2024.

**Strengths:**

+ The integration of stochastic prompt learning, deep prompting, and multiple prompts per class represents a novel contribution. As shown in Table 1, the proposed approach outperforms several state-of-the-art baselines.

**Weaknesses:**

- Although this integration has not been explicitly explored in prior work, all three underlying ideas have already been proposed in the existing literature. Therefore, the proposed approach appears somewhat straightforward.
- While the “k-shot all-class” setup (Lines 94-96) is interesting, the paper does not clearly explain why it is considered “closer to real-world deployment” than the widely used N-way k-shot (episodic) setting. Moreover, there seems to be no reason not to include comparisons between MIST and existing methods under the episodic setting as well.
- More recent methods should be included in the comparison. For example, although PromptMargin [1] (published in 2025) is cited in §2, its results are not reported in Table 1.
- Several ablation studies are missing. For instance, the effect of deep prompting and its configuration has not been evaluated. In addition, while the mean of the first prompt (i.e., ``A photo of a [CLS]'') is fixed throughout the experiments, it should also be tested as a learnable parameter.
- Bayesian prompt learning for vision–language models has already been explored in the literature (e.g., [2, 3]). Although the experimental settings differ, these works should be discussed to better contextualize and highlight the novelty of the proposed method.
- The listed contributions (Lines 56-65) are not orthogonal; at least points 1, 3, and 4 seem to overlap.
- While my understanding might be incomplete, it is unclear why the summation in Equation (5) (Lines 291-294) is taken over 1 to 2C. j takes only the values 1 or 2, doesn’t it?

[1] D. Brahma et al., Prompt Tuning Vision Language Models with Margin Regularizer for Few-Shot Learning under Distribution Shifts, in TMLR, 2025.

[2] M.M. Derakhshani et al., Bayesian Prompt Learning for Image-Language Model Generalization, in ICCV, 2023.

[3] X. Liu et al., Patch-Prompt Aligned Bayesian Prompt Tuning for Vision-Language Models, in UAI, 2024.

**Questions:**

Please refer to the Weakness section.

---

> ### Author Response · Authors · 2025-12-02
> **Response to Reviewer PyP4 (1/3)**
>
> We thank the reviewer for the insightful comments. We address all the reviewer questions below.
>
> >**While the “k-shot all-class” setup (Lines 94-96) is interesting, the paper does not clearly explain why it is considered “closer to real-world deployment” than the widely used N-way k-shot (episodic) setting. Moreover, there seems to be no reason not to include comparisons between MIST and existing methods under the episodic setting as well.**
>
> Early methods relied on the episodic meta-learning paradigm, with separate meta-training and meta-testing phases, where the adaptation was done over hundreds of episodes with fixed N-way k-shot support sets and Nq query set examples. Although, recent CLIP-based methods [1] discard the meta-training phase, and perform only meta-testing, this still involves fixed episodic sampling. This results in: (a) A different subset of N (typically 5) classes every episode, (b) Increased computational and time requirements due to thousands of episodic evaluation, (c) does not form a single classifier over the entire label space.
> In real-world scenarios (e.g., robotics, medical imaging, autonomous driving, etc.), such assumptions rarely hold, the model is deployed on all the target domain classes at once, and is evaluated on the entire test distribution. Our k-shot all class setting is designed to address this practical scenario: adaptation of a global single classifier on all classes and evaluation on the entire test distribution over the complete label space. While some recent CLIP-based finetuning methods adopt a similar all-class setting, they are limited to standard natural image benchmarks, with minimal domain and label semantic shifts. Our work addresses the more realistic paradigm which involves extreme domain and label shifts, which has been largely unexplored.
>
> The existing meta-learning-based methods typically consider fixed 5-way episodic classification, where the model is adapted to only $5\times k$ classes and tested on 15 samples from these 5 classes in each episode. This task is significantly easier compared to classification across the full set of categories, and generally results in inflated performance of the episodic-based methods. Comparing episodic 5-way accuracies to full-class accuracies (MIST setup) would therefore be misleading and not methodologically fair. For completeness, we consider a CLIP-based meta-testing work [1] and run it on our k-shot all class setup, and compare the results below. We consistently outperform PromptMargin across the different shots for all the datasets.
>
> | Shots   | Method        | EuroSAT | ISIC  | PlantDisease | ChestX | Avg |
> |---------|---------------|---------|-------|--------------|--------|------|
> | 1-shot  | PromptMargin  | 68.10 | 30.37 | 49.83 | 15.00    | 40.83 |
> |         | MIST          | 77.90   | 34.40  | 50.27        | 17.10   | 44.92 |
> ||||||||
> | 2-shots | PromptMargin  | 75.83   | 35.53 | 60.03        | 15.33  | 46.68 |
> |         | MIST          | 81.57   | 36.37 | 69.60         | 13.90   | 50.36 |
> ||||||||
> | 4-shots | PromptMargin  | 83.50    | 36.67 | 62.33        | 16.83  | 49.83 |
> |         | MIST          | 85.93   | 40.90  | 79.67        | 18.67  | 56.29 |
>
> [1] D. Brahma et al., Prompt Tuning Vision Language Models with Margin Regularizer for Few-Shot Learning under Distribution Shifts, in TMLR, 2025.
>
> >**More recent methods should be included in the comparison. For example, although PromptMargin [1] (published in 2025) is cited in §2, its results are not reported in Table 1.**
>
> We have provided the comparison of our proposed MIST with PromptMargin (TMLR 2025) above. Here we further provide comparisons with another recent CLIP-based few-shot method (2SFS) [2]. As observed, MIST consistently outperforms across all the settings.
>
> [2] Matteo, et. al., "Rethinking Few-Shot Adaptation of Vision-Language Models in Two Stages", in CVPR 2025.

---

> ### Author Response · Authors · 2025-12-02
> **Response to Reviewer Pyp4 (2/3)**
>
> | Shots     | Method       | EuroSAT | ISIC  | PlantDisease | ChestX | Average |
> |-----------|--------------|---------|-------|--------------|--------|---------|
> | 1-shot    | 2SFS (CVPR 2025)         | 73.7    | 26.23 | 43.9         | 13.43  | 39.32  |
> |           | MIST         | 77.9    | 34.4  | 50.27        | 17.1   | 44.92 |
> |           |              |         |       |              |        |         |
> | 2-shots   | 2SFS (CVPR 2025)         | 80.44   | 32.67 | 57.97        | 14.74  | 46.46  |
> |           | MIST         | 81.57   | 36.37 | 69.6         | 13.9   | 50.36   |
> |           |              |         |       |              |        |         |
> | 4-shots   | 2SFS (CVPR 2025)         | 85.5    | 36.62 | 68.96        | 19.05  | 52.53 |
> |           | MIST         | 85.93   | 40.9  | 79.67        | 18.67  | 56.29 |
> |           |              |         |       |              |        |         |
> | 8-shots   | 2SFS         | 89.1    | 44.91 | 81.51        | 17.73  | 58.31 |
> |           | MIST         | 88.63   | 52.7  | 87.47        | 16.5   | 61.33  |
> |           |              |         |       |              |        |         |
> | 16-shots  | 2SFS (CVPR 2025)         | 92.4    | 57.83 | 87.96        | 22.01  | 65.05   |
> |           | MIST         | 93.57   | 60.3  | 91.73        | 14.77  | 65.09 |
>
> >**Several ablation studies are missing. For instance, the effect of deep prompting and its configuration has not been evaluated. In addition, while the mean of the first prompt (i.e., ``A photo of a [CLS]'') is fixed throughout the experiments, it should also be tested as a learnable parameter.**
>
> As suggested, we provide the ablation results below (Table A). For MIST, we kept a fixed mean ("A photo of") for the first prompt, while keeping the second prompt learnable. Here, we make means for both the prompts learnable and report the results below. We observe that for lower shots, MIST comfortably outperforms this approach, while the opposite pattern is observed as the number of shots increase. This indicates that more number of shots is essential for learning both the prompt distributions together, consistent with what we described in L244-248 of our paper.
>
> | Shots   | Method | EuroSAT | ISIC  |
> |---------|--------|---------|-------|
> | 1-shot  | both   | 77.2    | 28.8  |
> |         | MIST   | 77.9    | 34.4  |
> |         |        |         |       |
> | 2-shots | both   | 81.7    | 34.83 |
> |         | MIST   | 81.57   | 36.37 |
> |         |        |         |       |
> | 4-shots | both   | 86.77   | 42.17 |
> |         | MIST   | 85.93   | 40.9  |
>
> Table A: Ablation study: Effect of learnable mean and variance for both the prompts (denoted by both).
>
> The effect of prompt depth is illustrated in Table B. The experiments are done for the 1-shot case. We observe that generally, the performance increases with increase in the prompt depth. However, after a certain point, the accuracy starts decreasing due to possible overfitting from increased number of parameters. This was also observed in [3,4]. Following these prior works, we have set the depth as 9 in our paper.
>
> | Method        | Eurosat | ISIC  |
> |---------------|---------|-------|
> | layers 1–5    | 72.17   | 23.57 |
> | layers 1–7    | 77.63   | 22.47 |
> | layers 1–9    | 77.90   | 34.40 |
> | layers 1–10   | 81.47   | 32.43 |
> | layers 1–12   | 79.47   | 28.07 |
>
> Table B: Effect of prompt depth.
>
> [3] Jia et. al., "Visual Prompt Tuning", in ECCV 2024.
> [4] Khattak et. al., "MaPLe: Multi-modal Prompt Learning", in CVPR 2023.
>
> >**The listed contributions (Lines 56-65) are not orthogonal; at least points 1, 3, and 4 seem to overlap.**
>
> We thank the reviewer for pointing this out. We agree that some points can appear related, so we clarify the distinct points here:
>
> 1. Problem and setting: We consider a realistic setting where extreme domain and label semantic shifts are present in an all-class k-shot setting. This sets the stage for how our considered setup is different from previous works.
>
> 2. Analysis: We analyze limitations of existing prompt tuning methods under severe distribution shifts, identifying gaps in current approaches.
>
> 3. and 4. MIST framework: We describe the different components of the proposed method. While multiple-classwise prompts help model the fragmented visual distributions, the Gaussian modeling helps to address overfitting. These two are orthogonal technical contributions. For better readability we can merge these two points.
>
> 5) This point highlights the empirical validation of our proposed approach with other state-of-the-art methods across several benchmarks.

---

> ### Author Response · Authors · 2025-12-02
> **Response to Reviewer Pyp4 (3/3)**
>
> >**Bayesian prompt learning for vision–language models has already been explored in the literature (e.g., [2, 3]). Although the experimental settings differ, these works should be discussed to better contextualize and highlight the novelty of the proposed method.**
>
> We thank the reviewer for pointing out these papers.
> [2] proposes to add a learnable Gaussian noise to each token inside the prompt to perform prompt tuning and mainly extends CoOp and CoCoOp. In contrast, [3] hierarchically generates prompt weights using generative modeling to introduce uncertainty in the prompt space, followed by minimizing a statistical distance between visual and text prompts for regularization. Moreover, these methods are not designed for addressing few-shot domain-shifted scenarios.
> Both the motivation and technical contributions of these works differ from our proposed MIST. MIST models multiple class-specific prompts distinctly as learnable Gaussian distributions, capturing multimodal visual feature distributions for realistic few-shot all-class adaptation under domain and label shifts.
>
> We have already discussed [2] in the paper (L84-87) and will add clarification regarding [3].
>
> >**While my understanding might be incomplete, it is unclear why the summation in Equation (5) (Lines 291-294) is taken over 1 to 2C. j takes only the values 1 or 2, doesn’t it?**
>
> We clarify the misunderstanding here. In Eqn. 4 (paper), $i^*$ selects the most similar prompt for the current class of the given image, i.e., the argmax over the two prompts for that class. The denominator in Eqn.5 (paper), however, sums over all the $2C$ prompts across all the classes (since each class has 2 prompts), and is effectively computing a softmax over all the possible prompts.
>
> >**Strength: The integration of stochastic prompt learning, deep prompting, and multiple prompts per class represents a novel contribution. Weakness: Although this integration has not been explicitly explored in prior work, all three underlying ideas have already been proposed in the existing literature. Therefore, the proposed approach appears somewhat straightforward.**
>
> We thank the reviewer for appreciating the novelty of integrating stochastic prompt learning, deep prompting, and multiple prompts per class. While we are indeed inspired by prior seminal works, our method is not a straightforward combination of existing ideas. The integration is specifically motivated by the unique challenges of this problem setup (few-shot adaptation under extreme domain and label semantic shifts), and the gaps identified in prior methods.
>
> Each component in MIST plays a complementary role (Sec. 4.2 of paper):
> The multiple class-specific prompts aim to capture the multimodal visual feature distributions that emerge under significant domain shifts.
> Each of these prompts are then stochastically modeled, with distinct parametric forms, following the observations in Sec. 4.1 (paper). This helps to mitigate overfitting, which can arise from the all-class few-shot adaptation setup.
>
> Importantly, the interaction between these components is non-trivial and naive combinations do not result in improvements (shown in ablations). To the best of our knowledge, no prior work has unified these components or evaluated such a design for few-shot adaptation under significant domain shifts.

---

### Meta-Review · Area_Chair_B7Hc · 2026-01-02

**Summary:**

Three experts in the field reviewed the paper and provided consistently negative ratings. The main concerns can be summarized as follows: (i) limited novelty due to the integration of existing components [Pyp4, p6n6]; (ii) limited experimental validation, including datasets, baselines, settings, and ablations [Pyp4, p6n6]; (iii) weak motivation for using multiple prompts to address extreme domain shift [p6n6, Pyp4]; and (iv) an unclear problem setting [QjNq, Pyp4].

Although the authors did not upload a revised manuscript, the AC carefully reviewed the original paper, the reviews, and the author responses. The AC agrees with [QjNq, Pyp4] that the proposed setting requires further clarification. First, the authors’ claim that the proposed “all-class K-shot” setting is more practical than the conventional N-way K-shot setting is overstated. In real-world deployments, not all scenarios require recognition across a large number of categories. For example, in certain environments, only a small number of wildlife categories may be present and need to be recognized. Moreover, collecting even a small number of samples for irrelevant or undesired categories may be impractical. This is precisely why the N-way K-shot setting remains meaningful. The “all-class K-shot” setting described in this paper can be viewed as a special case of N-way K-shot during meta-testing, with a larger number of classes. Although the authors argue that extensive episodic learning is performed during meta-testing, such training remains feasible and valid. For clearer differentiation, the authors should further explore evaluations under the standard N-way K-shot setting, potentially including a meta-training phase under N-way K-shot while evaluating under the all-class K-shot setting.

Furthermore, the AC also agrees with [p6n6, Pyp4] regarding the weak motivation for using multiple (specifically, two) prompts. The motivation and explanation are insufficiently clear, as the number of prompts appears to function largely as a tunable hyperparameter. As shown in Figure 6, model performance fluctuates substantially with different numbers of prompts. In particular, when using a single prompt, the model exhibits degraded performance compared to 2SFS and PromptMargin, which undermines the effectiveness of the proposed approach. In addition, the choice of using one fixed prompt and one learnable prompt appears ad hoc. The results suggest that with more shots, the model performs better when both prompts are learnable, likely because additional learning capacity is introduced as more data become available, as noted by the authors. This raises a natural question: under an N-way K-shot setting, where the number of classes is smaller and the learning requirement is reduced, could the model achieve superior performance using only a single prompt? From Figure 6, with 1-shot setting, with one prompt does not perform better compared with more prompt. It also conflicts with the author’s statement.

Overall, the paper, in its current form, is not ready for publication. The authors are encouraged to substantially revise the work and consider resubmission.

**Reviewer Concerns:**

In their response, the authors provided additional experiments involving more datasets, baselines, settings, and ablations to support their claims. As a result, the AC considers concern (ii) to be adequately addressed. However, the remaining major concerns regarding novelty, methodology (i.e., the use of multiple prompts), and the unclear problem setting remain unresolved.

**Reviewer Scores:**

Based on the author rebuttal, the AC believes that the major concerns have not been resolved. Therefore, all three reviewers are likely to maintain negative ratings.

---

### Decision · Program_Chairs · 2026-01-26

Reject